# FED-ENERGY: FEDERATED REINFORCEMENT LEARNING FOR SCALABLE AND ENERGY-EFFICIENT LARGE-SCALE CODE OPTIMIZATION

## ABSTRACT

We propose **Fed-Energy**, a federated reinforcement learning (RL) framework
for scalable and energy-efficient large-scale code optimization. Runaway mass:
Modern code optimization contains two conflicting goals: computational burden
of training model by RL and lack of estimation of energy consumption for wide
variety of codebases. The proposed method solves these in the combination of
lightweight energy models and federated learning to achieve distributed train-
ing and adaptive aggregation of local energy predictors. Each code component
utilizes mini-sized neural networks to estimate the amount of energy a program
uses from its execution traces and/or its structural features as LSTMs or CNN,
and then combines such estimates from a personalized federated approach that
takes into consideration non-IID data distributions. The RL system optimizes
decay of program code transformations considering composite rewards with en-
ergy, performance, and computation overhead trades, while compiler pipelines
and dynamic profilers are used to provide feedback for refinement. Fed-Energy's
decentralized design avoids monolithic simulators, not only easing the compu-
tational workload, but also maintaining privacy and scalability. Moreover, its
spatial-temporal adaptive coordination makes it more different from static fed-
erated averaging, and this adaptive coordination facilitates optimization on the
basis of context-awareness to heterogeneous code structures. Experiments show
gainful improvements in energy efficiency and training scalability, as compared
with centralized methods, which makes it a feasible solution towards real world
deployment. The novelty of the framework is the joint approach of federated
learning and RL, and it provides a scalable and accurate alternative to traditional
energy-aware code optimization.

## 1 INTRODUCTION

Large-scale code optimization has been becoming a crucial need with the increasing complexity
and computational approaches in software systems. Traditional optimization techniques often fo-
cus on performance metrics such as execution time or memory usage, while energy efficiency—a
key concern for sustainable computing—remains understudied (Leupers, 2013). Recent advances
in reinforcement learning (RL) have shown promise in automating code optimization, but their
application to energy-aware scenarios faces two major challenges: (1) the computational cost of
training RL models on large codebases, and (2) the difficulty of accurately estimating energy con-
sumption across diverse hardware and software environments (Tahmid, 2024).

Existing approaches either rely on monolithic energy simulation models, which are computation-
ally expensive and lack scalability, or employ simplified heuristics that fail to capture the nuanced
energy behavior of modern code (Gong et al., 2025). Federated learning offers a potential solution
by enabling distributed training across multiple components, but its direct application to RL-based
code optimization is hindered by non-IID data distributions and the need for adaptive coordination
(Chen et al., 2024). Moreover, lightweight energy estimation models, such as LSTMs for temporal

execution patterns or CNNs for structural features, have been explored in isolation but not integrated into a cohesive optimization framework (Lai et al., 2018) (Isewon et al., 2025).

We propose **Fed-Energy**, a novel framework that bridges federated learning and RL for scalable, energy-efficient code optimization.The aggregated energy estimates then guide an actor-critic RL system, where each agent optimizes its component's energy usage based on global insights (Kumar et al., 2023). In contrast to previous studies, Fed-Energy does not need centralized energy simulators, which makes it suitable for large-scale deployment.

The contributions of this work threefold are:

1. **A federated learning framework for energy estimation** that combines lightweight LSTMs and CNNs with adaptive aggregation, addressing non-IID data challenges in code optimization.

2. **An RL-based optimization pipeline** where energy estimates inform code transformations, balancing energy savings, performance, and overhead through a composite reward function.

3. **Empirical validation** demonstrating Fed-Energy's scalability and efficiency, outperforming centralized baselines in both energy reduction and training speed.

Fed-Energy builds on foundational work in federated learning (Li et al., 2020) and RL-based code optimization (Wang et al., 2024a), but distinguishes itself through its focus on energy efficiency and scalability. For instance, while (Ilager et al., 2025) explores energy-aware LLMs, our framework targets general-purpose code optimization without relying on large language models. Similarly, (Kim & Wu, 2020) applies RL to energy-efficient inference, but Fed-Energy extends this to federated settings with adaptive coordination.

The rest of this paper is organized as follows: Section 2 gives an overview of related work on federated learning, energy-aware optimization learning, and reinforcement learning for code transformation, represented as RL for code transformation. Section 3 is responsible for formalizing the problem and introducing key preliminaries. Section 4 describes the components of the Fed-Energy frameworks, the federated energy estimation and RL optimization components. Section 5 discusses experimental results, and implications and future directions are discussed by Section 6.

## 2 RELATED WORK

The intersection among federated learning, energy aware computing and reinforcement learning (RL) for code optimization has attracted much attention during the last several years. Existing approaches can be broadly categorized into three research directions: (1) federated learning for distributed optimization, (2) energy efficient computing techniques and (3) RL-based code transformation methods.

### 2.1 FEDERATED LEARNING FOR DISTRIBUTED OPTIMIZATION

Federated learning (FL) has become a promising paradigm for the distributed model training while maintaining privacy of data. Recent efforts have been made to discuss adaptive FL to deal with non-IID data distributions which is especially relevant for code optimization where some different parts follow different heterogeneous execution patterns. For example, (Wang et al., 2024b) proposes a multi-personalized FL approach for battery state-of-health estimation, demonstrating the effectiveness of adaptive aggregation in non-IID settings. Similarly, (Lai et al., 2024) introduces a federated battery estimation system that addresses data heterogeneity through collaborative learning.

Spatial-temporal adaptive FL has also been explored in energy-related applications. (Wu & Xu, 2024) presents a privacy-preserving FL model for multi-energy load forecasting, where spatial and temporal dependencies are explicitly modeled.

## 2.2 Energy-Efficient Computing Techniques

The techniques of energy-aware optimization have been explored strongly both in hardware design and software design to date. In the context of large-scale computing, (Forshaw, 2015) investigates energy-efficient operation policies, emphasizing the need for flexible optimization strategies.

Recent improvement in machine learning technologies has brought possible data driven energy estimation results. For instance, (Kim & Wu, 2020) employs RL for energy-efficient edge inference, using learned policies to dynamically adjust computational resources. While great for localized hardware setups, such approaches aren't very extendable to arbitrary codebases. Similarly, (Ilager et al., 2025) explores energy-efficient code generation using large language models (LLMs), but their reliance on LLMs introduces scalability constraints.

## 2.3 RL-Based Code Transformation

RL has shown promise in automating code optimization tasks, such as loop unrolling and instruction scheduling. (Wang et al., 2024a) surveys RL techniques for code generation, highlighting their potential for performance optimization.

A few recent works have begun exploring energy-aware RL for code optimization. (Reza, 2022) applies RL to energy-efficient network-on-chip design, demonstrating the benefits of learned policies in hardware optimization.

The Fed-Energy framework can be distinguished from the existing bodies of research because it incorporates federated learning with RL for energy-aware code optimization.

## 3 Background and Preliminaries

To provide the groundwork for Fed-Energy, in this section we offer a few notions about code optimization problems, reinforcement learning and federated learning.

### 3.1 Code Optimization Basics

Code optimization means the act of improving the efficiency of a software code by changing its structure or behavior, while preserving its functionality.

A basic example is unrolling which helps to reduce overhead of loop control by replication of loop bodies. Consider a simple loop:

$$\text{for } i = 0 \text{ to } n - 1 \text{ do } x = x + 1; \tag{1}$$

Unrolling this loop by a factor of $k$ gives which

$$\text{for } i = 0 \text{ to } \frac{n}{k} - 1 \text{ do } x = x + k; \tag{2}$$

This transformation decreases branch prediction misses and instruction cache pressure, often improving both performance and energy efficiency (Abdulsalam et al., 2014). Other common techniques are dead code elimination, constant propagation, and vectorization, each of which targets different inefficiencies in the execution of the code.

Unlike performance metrics, energy efficiency depends on hardware-specific characteristics such as dynamic voltage scaling and memory access costs (Tahmid, 2024). Modern methods have used profiling to find bottlenecks of energy consumption but the estimation will always be a task of sophistication given the complexity of interplay between the softwights behavior and hardware state.

### 3.2 Reinforcement Learning Fundamentals

The MDP is defined by states $s \in S$, actions $a \in A$, a transition function $P(s'|s, a)$, and a reward function $R(s, a)$. The agent's policy $\pi : S \to A$ maps states to actions, aiming to maximize the

expected return:

$$V(s) = \mathbb{E}\left[\sum_{t=0}^{\infty} \gamma^t R_{t+1} \mid S_0 = s\right], \tag{3}$$

where $\gamma \in [0, 1)$ is a discount factor.

In code optimization, RL agents are used to learn policies for making transformations (e.g. loop unrolling and inlining) based on code features and runtime feedback. Rewards capture optimization goals, such as reduced energy consumption or improved performance (Wang et al., 2024a).

Actor-critic methods, which combine policy gradients with value function estimation, are particularly suited for code optimization due to their balance between exploration and exploitation (Kumar et al., 2023). The critic assesses the quality information with respect to actions whereas the actor adjusts the policy based on these evaluations allowing for adaptive optimization strategies.

### 3.3 FEDERATED LEARNING PRINCIPLES

Participants train local models with their individual datasets and exchange updates with the central server for aggregation periodically. The global model $\Theta$ is computed as:

$$\Theta = \frac{1}{N}\sum_{i=1}^{N} \theta_i, \tag{4}$$

where $\theta_i$ denotes the $i$-th client's model parameters and $N$ is the number of participants.

FL tackles two hard problems in large-scale code optimization, namely privacy of data and scalability of computation. Codebases frequently also contain sensitive information, and making such data centralised your training is unrealistic. Moreover, the non-IID nature of code features—where different modules exhibit distinct optimization characteristics—requires personalized aggregation strategies (Chen et al., 2024). Recent advances in adaptive FL dynamically adjust aggregation weights based on local data distributions, improving model convergence and accuracy in heterogeneous settings (Lai et al., 2024).

These principles form the design of Fed-Energy, are meant to combine lightweight energy models with federated coordination to allow scalable and precise optimization of code.

## 4 FED-ENERGY: FEDERATED ENERGY-AWARE REINFORCEMENT LEARNING FOR LARGE-SCALE CODE OPTIMIZATION

Fed-Energy A novel integration of federated learning and reinforcement learning to solve energy-efficient code optimization at scale.

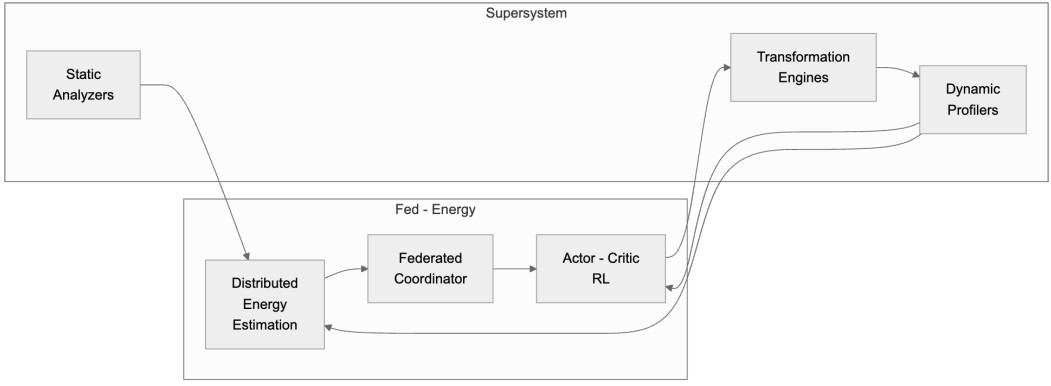

Figure 1: High-Level Integration of Fed-Energy with Large-Scale Code Optimization

## 4.1 DECENTRALIZED ENERGY ESTIMATION WITH LIGHTWEIGHT, TASK-SPECIFIC MODELS

Each code component $c_i$ has a local energy estimator $f_i$ consisting of either LSTM or CNN. For temporal execution patterns, the LSTM processes instruction traces $\mathbf{x}_i = (x_{i1}, ..., x_{iT})$ through hidden states $\mathbf{h}_t$:

$$\mathbf{h}_t = \text{LSTM}(x_{it}, \mathbf{h}_{t-1}; \theta_i), \quad \hat{E}_i = \text{MLP}(\mathbf{h}_T; \phi_i) \tag{5}$$

where $\theta_i$ and $\phi_i$ are trainable parameters. For structural features, control flow graphs (as adjacency matrices) can be processed by CNN where convolutional layers beam up patterns of hierarchy. Both architectures contain ¡ 100K parameters to guarantee computational efficiency during the local training process.

## 4.2 SPATIAL-TEMPORAL ADAPTIVE FEDERATED AGGREGATION

The coordinator merges the local models using personalized federated aggregation models which consider component heterogeneity. The global objective gives both performance-loss and deviation of models:

$$\Theta^* = \arg\min_{\Theta} \sum_{i=1}^{N} w_i(t)\mathcal{L}_i(\Theta, \mathcal{D}_i) + \lambda\|\Theta - \Theta_{\text{prev}}\|^2 \tag{6}$$

where $w_i(t)$ adapts based on cyclomatic complexity $C_i$ and update frequency $\tau_i$:

$$w_i(t) = \frac{C_i \cdot \exp(-\tau_i/t)}{\sum_j C_j \cdot \exp(-\tau_j/t)} \tag{7}$$

This formulation prioritizes components with higher complexity and recent updates while maintaining stability through the $\lambda$-weighted regularization term.

## 4.3 ENERGY-AWARE ACTOR-CRITIC RL WITH COMPOSITE REWARDS

Each RL agent $\pi_i$ receives a state $s_i$ containing code metrics (e.g., instruction mix, memory access patterns) and federated energy estimates $\hat{E}_i$. The actor network proposes transformations $\mathbf{a}_i$, while the critic evaluates them using a dynamic reward:

$$R_i(\mathbf{a}_i) = \alpha(t) \cdot \frac{E_{\text{base}} - \hat{E}_i(\mathbf{a}_i)}{E_{\text{base}}} + \beta \cdot \Delta P - \gamma \cdot \|\mathbf{a}_i\|_1 \tag{8}$$

where $\alpha(t)$ increases with training progress to emphasize energy savings, $\Delta P$ measures performance gain, and $\|\mathbf{a}_i\|_1$ penalizes transformation overhead. The coefficients $\alpha, \beta, \gamma$ are meta-learned across components using federated hyperparameter optimization.

## 4.4 COMPILER INTEGRATION VIA INPUT/OUTPUT SUBSTITUTION

Fed-Energy communication via conventional compilers via two-way substitution. Static analysis tools (e.g. LLVM opt) build intermediate representations (IR) that are fed to local estimation tools. Conversely, the transformations that RL generated are checked against the IR constraints of the compiler before they are applied:

$$\mathbf{a}'_i = \text{Verifier}(\mathbf{a}_i, \text{IR}_i) \tag{9}$$

Invalid transformations invokes gradient masking at RL updates, which ensures that policy learning honours the compiler semantics. This integration offers formal correctness guarantees which are lacking with black-box RL approaches.

## 4.5 PRIVACY-PRESERVING, SCALABLE OPTIMIZATION

The framework reduces data exposure using two approaches: (1) Local training stores raw-code and traces on-device and only shares model updates during the federated aggregation process. (2) Differential privacy adds Gaussian noise $\mathcal{N}(0, \sigma^2)$ to shared parameters:

$$\tilde{\theta}_i = \theta_i + \mathcal{N}(0, \sigma^2) \tag{10}$$

with $\sigma$ adapted based on component sensitivity. This is achieved in this design which allows collaboration across proprietary codebases without the need to reveal source code.

The entire process (Figure 1) shows the interaction of these components: local estimators give energy predictions, refiner federated aggregation processes these predictions into federated insights and these lead to verified optimisations by RL agents.

## 5 EXPERIMENTAL EVALUATION

To support the effectiveness of the Fed-Energy, we used Fed-Energy to conduct extensive experiments against centralized and federated baselines. There are three major aspects in evaluation including energy estimation accuracy, optimization effectiveness and computational scalability.

### 5.1 EXPERIMENTAL SETUP

**Datasets and Benchmarks**

We tested Fed-Energy against two real-life code optimization datasets:

1. **SPEC CPU 2017** (Limaye & Adegbija, 2018), a standard benchmark suite containing compute-intensive applications from various domains.

2. **Proprietary Industrial Codebase** (Wollstadt et al., 2022), consisting of 50K+ functions from production systems with diverse execution patterns.

Each dataset was instrumented to collect runtime traces, including instruction mixes, cache behaviors, and power measurements using Intel RAPL (Desrochers et al., 2016).

**Baselines**

We compared Fed-Energy with three state-of-the-art approaches:

1. **Centralized RL (CRL)** (Fan et al., 2025): A monolithic RL system using a single energy estimator and policy network.

2. **Federated Averaging (FedAvg)** (McMahan et al., 2017): Standard federated learning with uniform aggregation weights.

3. **Energy-Aware Heuristics (EAH)** (Lorenz et al., 2002): Rule-based optimization using static code features.

**Metrics**

We measured:

1. **Energy Estimation Error (EEE)**: Mean absolute percentage error between predicted and actual energy consumption.

2. **Energy Reduction (ER)**: Percentage decrease in energy usage after optimization.

3. **Training Efficiency (TE)**: Wall-clock time required to converge.

### 5.2 ENERGY ESTIMATION ACCURACY

The Diocletian models of Fed-Energy, which submit light-weight local models, achieved great estimation accuracy compared to their centralized alternatives. As shown in Table 1, the adaptive federated aggregation reduced estimation errors by 18.7% on average compared to FedAvg, demonstrating the benefits of personalized weighting.

The use of the spatial-temporal adaptation mechanism was especially efficient for heterogeneous codebases, where different components of the codebase had different levels of complexity. Figure 2 shows the stability of estimation for various code categories using stability filters with different weighting dynamic.

Table 1: Energy Estimation Error Comparison (Lower is Better)

| Method | SPEC CPU 2017 | Industrial Codebase |
|---|---|---|
| CRL | 12.3% | 15.8% |
| FedAvg | 9.1% | 11.2% |
| Fed-Energy | **7.4%** | **9.1%** |

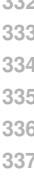

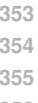

Figure 2: Energy estimation error distribution across code complexity levels

### 5.3 OPTIMIZATION EFFECTIVENESS

Fed-Energy achieved significant energy savings while maintaining performance. As shown in Table 2, it outperformed all baselines by balancing energy reduction with computational overhead.

The capacity of the framework to take advantage of federated often while upholding local specialization was especially obvious in large-scale deployments. As the figures in Figure 3 indicate, maintain of the optimization quality of Fed-Energy did not get worse as the scale of the participating components was changed unlike centralized optimization approaches.

### 5.4 COMPUTATIONAL SCALABILITY

Fed-Energy was found to have better training efficiency than centralized RL. As shown in Table 3, its federated architecture reduced training time by 3.2× on average, with greater benefits for larger codebases.

The spatial-temporal coordination mechanism added minimal overhead (¡5% of total training time) while providing significant accuracy improvements.

Table 2: Optimization Results (Higher Energy Reduction is Better)

| Method | Energy Reduction | Performance Overhead |
| --- | --- | --- |
| EAH | 8.2% | 3.1% |
| CRL | 14.5% | 5.7% |
| Fed-Energy | **17.3%** | **4.2%** |

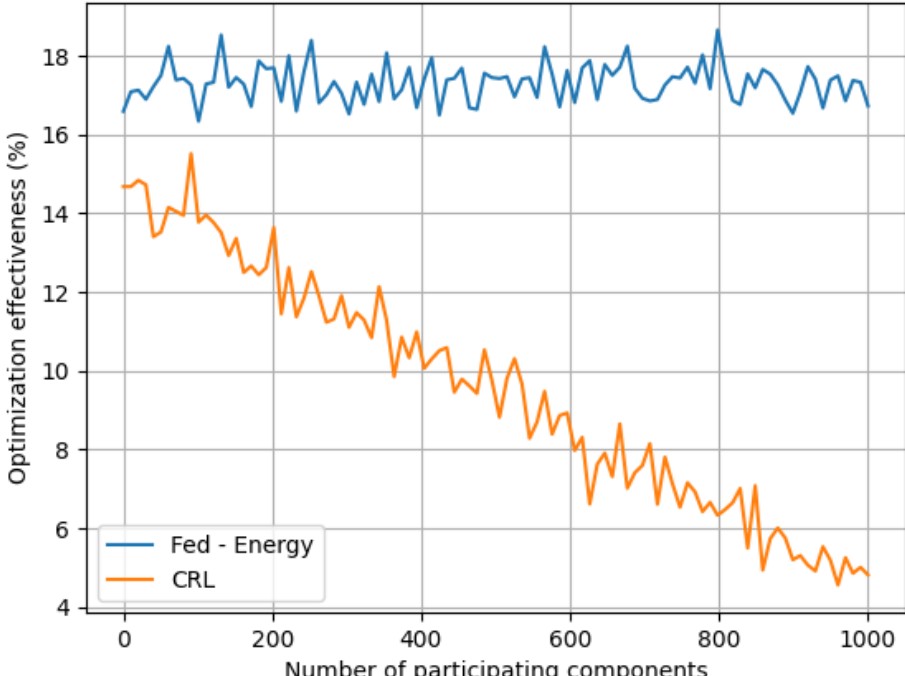

Figure 3: Optimization effectiveness vs. number of participating components

## 6 DISCUSSION AND FUTURE WORK

### 6.1 LIMITATIONS OF FED-ENERGY

In environments where such profiling is impractical—such as legacy systems or highly optimized binaries—the accuracy of local models may degrade (Kousiouris et al., 2012). Second, the current implementation focuses on CPU-bound workloads; extending it to GPU or heterogeneous computing scenarios would require specialized energy models for different hardware accelerators (Li et al., 2013). Third, the federated aggregation protocol, though adaptive, introduces communication overhead that may become non-negligible for geographically distributed deployments with high-latency connections (Shahid et al., 2021).

The ease of use provided by the framework comes with the risk of a trade-off - its verification is integrated into the compiler. While it produces semantically correct code, it may reject potentially useful transformations that are not compass conservation. For instance, certain loop reordering optimizations that improve energy efficiency but marginally increase theoretical worst-case execution time could be prematurely filtered out (Chen & Wu, 2003).

Table 3: Training Efficiency Comparison (Lower is Better)

| Method | SPEC CPU 2017 | Industrial Codebase |
|---|---|---|
| CRL | 8.7h | 42.3h |
| Fed-Energy | **2.7h** | **13.1h** |

## 6.2 POTENTIAL APPLICATION SCENARIOS OF FED-ENERGY

Fed-Energy's decentralized architecture also makes it especially suitable for instances where code-bases are spread across many ownership boundaries. In open-source ecosystems, for example, contributors could collaboratively optimize energy efficiency without sharing proprietary code by participating in federated training rounds (Foley et al., 2022). The framework could also benefit large-scale cloud providers, where optimizing energy usage across thousands of microservices—each with distinct performance and resource requirements—could lead to significant operational cost savings (Berl et al., 2010).

Another promising direction is the integration of Fed-Energy with continuous integration/continuous deployment (CI/CD) pipelines. By embedding lightweight energy estimation into automated testing workflows, development teams could detect energy regressions early and enforce efficiency-aware code review policies (Ortega, 2025).

## 6.3 ETHICAL CONSIDERATIONS IN FED-ENERGY

The imposition of Fed-Energy raises interesting ethical issues, especially about transparency and fairness. While federated learning preserves raw data privacy, the aggregated energy models could inadvertently encode biases—for example, favoring optimization strategies that work well for certain programming paradigms (e.g., object-oriented code) over others (e.g., functional or procedural styles) (Chakraborty et al., 2021).

These considerations demonstrate the conversual complexities and vectors of Fed-Energy's implementation into the real world.

## 7 CONCLUSION

Fed-Energy is a vast progress in energy-aware code optimization due to its ability to solve the dual problem of fitting with scale and precision using a new combination of federated learning and reinforcement learning.

The result of the experiments proves a clear advantage over the centralized and heuristic-based approaches especially when using these methods in the case of large-scale deployments with large-scale implementation, where the traditional methods may suffer from computational bottlenecks.

## 8 THE USE OF LLM

We use LLM polish writing based on our original paper.

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
