# OpenReview forum: "Fed-Energy: Federated Reinforcement Learning for Scalable and Energy-Efficient Large-Scale Code Optimization"
_ICLR.cc/2026/Conference — Submitted to ICLR 2026_

### Official Review · Reviewer_fN31 · 2025-10-25

**Soundness:** 2
**Presentation:** 2
**Contribution:** 2
**Rating:** 4
**Confidence:** 4

**Summary:**

The paper is about Fed-Energy. It addresses two key challenges in RL-based compiler optimization: (1) the high computational cost of centralized training and (2) the difficulty of accurately estimating energy usage across heterogeneous codebases and hardware. Fed-Energy uses lightweight models (LSTM/CNN) with federated learning, enabling distributed training without sharing raw code. It further incorporates an actor-critic RL system that uses a composite reward function balancing energy savings, performance gains, and computational overhead. The paper also shows spatial-temporal adaptive aggregation for federated coordination, and integrates compiler verification to ensure semantic correctness of code transformations.

**Strengths:**

The paper strengths are:

The spatial-temporal adaptive aggregation in federated training is an extension of standard FedAvg. This enables adaptation to heterogeneous code complexity and update frequency. Integrating compiler-level semantic verification with RL-based optimization is improves reliability, a significant contribution beyond purely black-box RL systems.

The framework’s privacy-preserving design and scalability make it applicable to distributed and industrial codebases in compiler optimizations. The empirical gains suggest that Fed-Energy could meaningfully addition to research in federated RL for systems optimization.

**Weaknesses:**

Weaknesses

The industrial dataset’s description isnt lacking. Adding details on its diversity and representativeness would improve the paper. The ablasion doesnt compare with other federated reinforcement learning frameworks or compiler-optimization-specific approaches: AutoPhase, AlphaDev-inspired RL baselines

The three-layered system (local estimators, federated coordinator, RL optimization loop) may introduce engineering complexity that could offset some of its claimed efficiency gains. A discussion on deployment overhead or resource footprint would be helpful. Therefore, an ablation study isolating the effects of spatial-temporal aggregation, differential privacy, and federated hyperparameter optimization would clarify their individual contributions.

**Questions:**

A few questions for the authors:

Adding more ablation breakdown to isolate the contribution of adaptive aggregation, differential privacy, and federated hyperparameter optimization, would improve the paper

Have the authors tested Fed-Energy’s scalability beyond SPEC? For example, open-source benchmarks like LLVM test suites?

How does Fed-Energy compare with other federated RL algorithms in terms of convergence speed, stability, or robustness?

**Details Of Ethics Concerns:**

-

---

### Official Review · Reviewer_GeWk · 2025-10-28

**Soundness:** 2
**Presentation:** 2
**Contribution:** 3
**Rating:** 2
**Confidence:** 3

**Summary:**

This paper proposes Fed-Energy, a reinforcement framework for large-scale code optimization. It aims to address two challenges. First, the computational cost of training RL models on large codebases. Second, estimating the correct energy consumption across different hardware environments. Fed-Energy instead integrates lightweight neural energy estimators with federated learning coordination, enabling distributed optimization that preserves privacy and scales effectively.

**Strengths:**

Originality: The paper presents a novel and compelling contribution by integrating federated learning and RL for energy-aware code optimization, a domain historically driven by heuristic or centralized methods. The proposed use of lightweight LSTM/CNN-based local energy estimators within an actor-critic RL framework represents an effective and creative fusion of model-based energy prediction with policy optimization. This combination meaningfully advances the state of the art in scalable and privacy-preserving code optimization.

Quality: The experimental design is well thought out, leveraging both standardized benchmarks (SPEC CPU 2017) and a large-scale proprietary industrial dataset comprising over 50,000 functions. The inclusion of diverse datasets enhances the credibility and generalizability of the findings. Furthermore, the paper provides meaningful comparisons with appropriate baselines, clearly demonstrating the advantages of the proposed approach over existing methods.

Clarity: The paper is clearly structured and generally well written.

Significance: This work holds strong potential for impact in both industry-scale software ecosystems and the broader field of sustainable computing.

**Weaknesses:**

1. The paper introduces several important mechanisms, but their individual contributions to the overall performance are not clearly isolated. An ablation study would significantly strengthen the paper by demonstrating how each component impacts energy estimation accuracy, optimization quality, and training efficiency.

2. The paper states that the adaptive coordination mechanism adds “5% of total training time,” but this claim is not supported by empirical evidence or quantitative analysis. Also,  the symbol “¡” appears to be a typo

3. The comparison with baselines such as FedAvg and EAH is somewhat outdated, given recent progress in federated and reinforcement learning optimization methods. The author should compare with more recent baselines.

**Questions:**

See the Weaknesses.

---

### Official Review · Reviewer_TQLK · 2025-10-31

**Soundness:** 2
**Presentation:** 1
**Contribution:** 2
**Rating:** 2
**Confidence:** 3

**Summary:**

The paper introduces Fed Energy, combining federated learning with actor–critic RL and compiler verification to optimize code for energy efficiency at scale. Experiments on SPEC CPU2017 report lower energy estimation error than FedAvg, higher energy savings than heuristic and centralized RL baselines, and faster training.

**Strengths:**

•	Reports promising gains in energy reduction and training time.

•	Evaluated on the standard SPEC CPU2017 benchmark.

**Weaknesses:**

•	I find the paper hard to read since the problem is not clear. The paper tries to solve many problems at the same time, which might confuse readers. It would be better to rewrite the paper to clarify exactly what are the problems being solved, why each problem is important to solve (and why a simpler solution is not enough), and how it is solved. As the paper is written now, it mixes all the problems together, making it hard for the reader to appreciate the contribution.

•	The authors use an “industrial codebase” as a benchmark and cite (Wollstadt et al., 2022), and say that it “consists of 50K+ functions from production systems with diverse execution patterns” but as far as I understand, the cited paper is a 3D geometry dataset which cannot serve as a source of “50K+ functions or program traces". Can the authors provide more clarity on these citations?

•	Reducing energy consumption through compiler methods is a well-studied area. There are many compiler methods specialized in optimizing energy consumption. Some existing work also minimizes energy consumption indirectly by optimizing code (minimizing execution time and improving data locality). Therefore, many existing classical code optimization methods also implicitly target energy minimization. The authors do not provide any comparison with existing compiler methods, though (whether those designed specifically for minimizing energy consumption or those designed for automatic code optimization in general). It is not clear how does the proposed method compare to classical compiler methods (whether those designed for energy optimization or those designed for code optimization in general). Important related work in this area is also missing in the paper. The paper does not talk about classical compiler methods for energy optimization. Examples of work to consider: “Energy-Aware Register Allocation for VLIW Processors”, “Identifying Compiler Options to Minimise Energy Consumption for Embedded Platforms”, “Instruction scheduling for power reduction in processor-based system design”, “Reducing instruction cache energy consumption using a compiler-based strategy”, “Compiler-level DMA-aware multi-objective dynamic SPM allocation”, “Bandwidth-Aware Loop Tiling for DMA-Supported Scratchpad Memory”.

•	Automatic code optimization using RL is also widely studied. The authors do not compare with any state-of-the-art RL-based method (e.g., MLGO which is deployed in LLVM, AutoPhase, or CompilerGym). While such a comparison is not necessary, it would help in showing the value of the proposed RL method compared to classical RL-based methods in compilers.

•	The paper uses many awkward expressions (“Runaway mass,” “Diocletian models,” “beam up patterns”). A careful language pass is needed.

**Questions:**

•	What is the actual industrial dataset? If proprietary, provide summaries and access conditions; if not, correct the citation.

•	Can you add more related work about classical compiler methods for energy optimization and code optimization (many mentioned above)?

---

### Meta-Review · Area_Chair_E93C · 2026-01-07

**Summary:**

Reviewers are concerned about the paper on Fed-Energy, a federated RL framework for energy-aware code optimization, because of its unclear problem framing, poor presentation, insufficient comparisons to prior compiler/RL methods, lack of ablation studies, and unsupported efficiency claims. These issues, combined with limited empirical depth, led to a reject recommendation.

**Reviewer Concerns:**

No rebuttal is provided so no concern is addressed properly.

**Reviewer Scores:**

- Reviewer TQLK: maintain 2 (fundamental issues with clarity, citations, and related work unaddressed).
- Reviewer GeWk: maintain 2 (no ablation or evidence for claims; outdated baselines).
- Reviewer fN31: maintain 4 (needs ablations, comparisons, and scalability details).

---

### Decision · Program_Chairs · 2026-01-26

Reject